# The Intra- and Inter-Regional Economic Effects of Smart Tourism City Seoul: Analysis Using an Input-Output Model

Hyunae Lee  and Sunyoung Hlee *

Smart Tourism Research Center, Kyung Hee University, 26 Kyungheedae-ro, Dongdaemun-gu, Seoul 02447, Korea; halee8601@khu.ac.kr
* Correspondence: onoonee@gmail.com; Tel.: +82-961-2353

**Abstract:** A competitive smart tourism city can be a solution for building resilience to address current and future crises and even be a booster of the economic effects of tourism, since it has an economic ripple effect both inside and outside of the city. This study tried to estimate the intra- and inter-regional economic effects of the smart tourism city Seoul, which has high competitiveness as a smart tourism city. First, this study tried to objectively clarify the scope of smart tourism based on a Delphi survey and then adopted the inter-regional input–output model. The results showed that smart tourism city Seoul is expected to create high income, high value-added, and job creation effects inside the city, and will greatly contribute to securing tax revenue. Outside of the city, smart tourism city Seoul is expected to induce high production effects. Based on these results, this study highlights the economic effects of a smart tourism city, which describes the convergence of technology and tourism.

**Keywords:** smart tourism city; Seoul; intra- and inter-regional economic effect; inter-regional input–output model



## 1. Introduction

Today's tourism cities are facing numerous issues resulting from tourism's vulnerability to unprecedented situations or crises, such as COVID-19. The tourism industry has built resilience in its own way in order to address economic, social, and environmental risks (e.g., terrorism, natural disasters, infection diseases, etc.), despite the inherent characteristics of being vulnerable to the external environment. However, COVID-19 has brought an unprecedented crisis to tourism [1,2]. Before COVID-19, tourism was used to generate enormous economic effects throughout regions and countries, and was even regarded as a means of poverty alleviation in developing countries [3]. However, the economic prosperity that tourism brought has vanished after travel was nearly completely banned subsequent to the pandemic declaration by World Health Organization (WHO).

In this situation, a smart tourism city, a combination of smart tourism and a smart city [4], is considered to be a solution for building resilience to address current and future crises, and even be a booster of the enormous economic effects of tourism [1,2,5]. One possible reason is that numerous economic entities of a smart tourism city are converged, and this convergence can provide a wide range of economic effects to the city and a spillover effect to nearby regions as well [6]. Convergence is the common characteristic of smart tourism and a smart city. Numerous platforms and stakeholders converge and form a smart tourism ecosystem [6,7]. This convergence makes the economic effects of a smart tourism city spread over a wide range, regionally and industrially. Therefore, securing competitiveness as a smart tourism city can be a solution for reopening tourism and the economy.

Thus, it is not surprising that numerous cities have tried to be smart tourism cities. Amsterdam, Singapore, Barcelona, New York City, Copenhagen, and Seoul have been regarded as leading smart tourism cities. Among them, Seoul is implementing an active smart tourism city policy led by the government. The Korean government has recently

kicked off a project to provide 4 billion KRW (about USD 3.3 million) to cities for creating a smart tourism city [8]. The president of South Korea expedited the establishment of smart tourism infrastructure and highlighted the smart tourism ecosystem as one of the tourism innovation strategies [9]. Moreover, Seoul is evaluated as one of the most competitive smart tourism cities with a high level of digital readiness and tour safety, following Singapore, Amsterdam, and New York City [10].

However, little attention has been paid to the intra- and inter-regional economic effects of smart tourism and its economic relationship with other industries, from the perspective of city scale. Although some previous studies have evaluated the economic effects of smart tourism [11–13], they focused on the impact of smart tourism as a whole national economy without distinguishing the intra- and inter-regional economic effects, or they depended on only the researchers' arbitrary decisions when they clarified the scope of smart tourism, which can cause problems of overestimation or underestimation.

Therefore, regarding these points, this study raises the following research questions:

RQ1: How much does a smart tourism city contribute to the economies of the city itself and its nearby regions?

RQ2: How does the smart tourism industry have economic relationships with other industries in a smart tourism city?

This study aims to answer these two research questions by adopting the inter-regional input–output (I–O) model to evaluate the intra- and inter-regional economic impacts of a smart tourism city, and by calculating the forward and backward linkage effects of smart tourism to understand its economic relationships with other industries in a smart tourism city.

Section 2 of the current study presents a review of the literature on smart tourism cities and the inter-regional I–O model. Section 3 explains the research design involving three steps: (1) defining the scope of smart tourism based on the Delphi survey, (2) analyzing and calculating the inducement coefficient values, and (3) estimating the economic effects. Section 4 reveals the results of the two-round Delphi survey, the intra- and inter-regional economic effects, and the linkage effects of smart tourism. Finally, Section 5 discusses the results and provides present theoretical and practical implications and direction for future studies.

## 2. Theoretical Background

### 2.1. Definition of a Smart Tourism City

With the explosive growth of technology, the ways by which global cities are constructed, consumed, and shared have changed and "smartized" [4,14]. The concept of the smart city has been defined by numerous earlier researchers [15–17], but the commonly stated attribute of a smart city is that it leads to efficiency improvement, sustainability, eco-friendliness, and improved resident/tourist quality of life/visit through connectivity via information communication technologies (ICTs) [4]. At the same time, tourism has been also technologically, economically, and socially developed and smartized with the convergence of ICT and tourism [18,19]. The concept of smart tourism has put emphasis on achieving a symbiotic relationship between tourists and citizens and creating economic and social value [18,19].

Therefore, as the term "smart" is added to cities and tourism, a smart tourism city is defined as an innovative and sustainable city that achieves economic and social values and enhances the city's competitiveness by collecting, analyzing, visualizing, and modeling real-time big data generated throughout the city and sharing it with all stakeholders of the smart tourism ecosystem [18–22]. By reviewing the estimation standards proposed by numerous organizations for assessing a smart tourism city, we can identify how a smart tourism city is perceived. The European Union (EU) has annually evaluated the outstanding smart tourism capitals with the following four categories: accessibility, sustainability, digitalization, and cultural heritage and creativity [23]. The Seoul Tourism Organization assessed 12 cities with the following 5 categories: attractiveness, accessibility, digitalization

readiness, sustainability, and collaborative partnership [4,10]. These categories reflect that a smart tourism city is one that has (1) enough infrastructure for anyone regardless of age, nationality, or physical ability, (2) balanced economic and social systems for a fair distribution of economic and social benefits to all stakeholders of the smart tourism ecosystem, (3) social efforts for environmental sustainability, (4) strategies for natural and cultural heritage by offering an innovative tourism experience based on information communication technologies (ICTs).

Therefore, we can state that a smart tourism city achieves economic, social, cultural, and environmental sustainability based on the convergence of tourism and technologies. Among these numerous aspects of sustainability achieved by a smart tourism city, this study focused on the economic one. Earlier scholars have highlighted the importance of economic sustainability of a smart tourism city by noting that only a city achieving sustainable economic growth, and travelers' and residents' high quality of visits and lives, based on technologies, can be regarded as a smart tourism city [14,24].

### 2.2. Economic Effects of a Smart Tourism City

A smart tourism city can evoke positive economic effects by increasing the economic benefits and reducing costs. In terms of increasing benefits, both tourism and technology, which are the two major fields of a smart tourism city, have already had positive economic effects. Tourism also has a broad effect because it brings numerous opportunities that build or upgrade Social Overhead Capital (SOC), like roads, highways, bridges, airports, and so on [25], which in turn brings widespread economic prosperity to the area around the tourist destination. In addition, technology is regarded as a core solution for overcoming risks resulting from COVID-19 by building tourism resilience [1,5]. For instance, self-service kiosks, Artificial Intelligence (AI) speakers, robot concierges, and so on, can minimize human contact between customers (tourists) and employees [26], which consequently contributes to limiting further spread of COVID-19. Moreover, immersive technologies (e.g., Virtual Reality (VR), Augmented Reality (AR), etc.) can increase prospective travelers' desires and expectations to visit the real destination and enhance the experience by serving as a substitute tourism experience [27]. In addition, residents of a smart tourism city may increase their knowledge management skills by being frequently exposed to technologies [13]. In terms of reducing costs, a smart tourism city offers new ways of managing city resources and tourist flows effectively [21]. A smart tourism city has control over information sources and flow between the various economic entities making up the city, and this power of control gives the city sustainable economic power [20] and productivity by reducing the cost of running the city [13].

In a smart tourism ecosystem, there are numerous stakeholders, and they are connected with each other [22]. In this vein, securing competitiveness as a smart tourism city does not simply result in the economic growth of that city, but also has positive economic ripple effects in other regions [12]. Therefore, this study tried to evaluate the economic effects of a smart tourism city within the city itself and other nearby regions by adopting the inter-regional I–O model.

### 2.3. Inter-Regional Input–Output Model

Diverse industries make up the national economy by buying, producing, and selling a variety of goods and services, through which raw or subsidiary materials are used for the production activities of other industries, consumed or invested within the country, or exported overseas [28]. These transaction details are arranged in a matrix called the "I–O table", which is published periodically; an analysis method that quantitatively grasps the inter-industry relationship by using this table is called the "input–output analysis" [28].

The inter-regional I–O model was originally developed by Isard [29], who noted the importance of considering the inequalities in the geographic distribution of population, income, and resources. Because the inter-regional I–O table consists of transaction information throughout regions and industries [28], it is useful for identifying the economic

structure of each region and their inter-regional and inter-industrial relationships. Therefore, it is also the proper method for estimating the economic effects of a smart tourism city, where numerous industries are converged.

In addition, the inter-regional I–O model reduces the risk of overestimating the economic effects by distinguishing between the economic ripple effects leaking to other regions and the effects within the region. Therefore, we can estimate the economic effects of the investment in smart tourism that is occurring within the city itself and in other regions.

Previous studies have adopted the inter-regional I–O model to estimate the economic effect of specific events [30], investments or funds [31,32], and specific industries [33,34] in a specific region and its nearby near regions (see Table 1). Lee et al. investigated the economic impact of a mega-event (2012 Yeosu Expo in Korea) on the host city (Yeosu) and its nearby regions [30]. Among the economic effects generated from the Expo, approximately 80% impacted the host city, and the other 20% leaked to the other regions. The economic effects of funds or investments, such as an Olympic-related investment [31] and the European Union (EU) structural funds [32], have been evaluated. These studies contributed to providing meaningful implications about the effectiveness of investments and policies. The impacts of numerous industries, such as convention and exhibition industries [32], on a smart tourism city [13], have been estimated as well. Because these studies estimated the economic impacts of relatively newer or convergence industries, they could provide meaningful implications for deciding future directions.

**Table 1.** Previous studies applied the I–O model.

| Category | Scholar | Purpose |
|---|---|---|
| Mega-event | Lee et al. (2017) | Estimation of economic effects of the 2012 Yeosu Expo in Korea on the host city and its nearby regions |
| Investment or fund | Zhang & Zhao (2007) | Estimation of the economic effects of the 2018 Beijing Olympic-related investments on Beijing, its surrounding areas, and the rest of China |
| | Pérez et al. (2009) | Estimation of the economic effects of the EU structural funds on Spanish regions |
| Specific industries | Lee et al. (2013) | Estimation of the economic effects of convention and exhibition business on Daegu (a city in South Korea) |
| | Lee et al. (2019) | Estimation of the economic effects of tourists' expenditure in smart tourism city Busan (a city in South Korea) |

## 3. Research Design

The analysis of this study involves three steps (see Figure 1). The first step involves conducting a two-round Delphi survey. Experts who are knowledgeable in smart tourism, smart cities, were involved. In the first round, 12 experts were asked to choose smart tourism-related industries among the 80 sectors reported in the inter-regional I–O table (two industries, mining products and the tobacco industry, which are not traded in Seoul, were excluded from the analysis). The industries selected by more than half of the experts were rated by the experts in the second round. The threshold was decided based on the narrow classification proposed by Jun et al. [34]. The experts were required to answer the degree of relationship between the concept of smart tourism and each selected industry on a 5-point Likert scale (1 = Strongly Not Related, 7 = Strongly Related). Industries with an average score of 4 or more were finally selected as the smart tourism industries. As this survey targeted smart tourism experts and professionals, we increased objectivity and expertise, and decreased possible risks caused by the researchers' arbitrary determination. In addition, although a considerable number of studies have focused on smart tourism,

more efforts are needed to clarify the definition and scope of smart tourism. Therefore, it was necessary to clarify the scope of what industries belong to smart tourism prior to estimating its economic effects.

| | |
|---|---|
| **Step 1:**<br>**Defining the scope of smart tourism** | **1. The first round of Delphi survey**<br>Smart tourism professionals are asked to choose industries relevant to smart tourism.<br>**2. The second round of Delphi survey**<br>Smart tourism professionals are asked to answer the degree of relation-ship between smart tourism and each selected industry on a 5-point Lik-ert scale (1=strongly not related, 7=strongly related)<br>**3. Confirming the scope of the smart tourism industry**<br>Industries that were evaluated by more than half of the experts as smart tourism related industries, and the degree of relatedness of 4 or more were selected as the smart tourism industries. |
| **Step 2:**<br>**Analyzing and calcu-lating inducement coefficient values** | **1. Construction of the input coefficient matrix and the product induce-ment coefficient matrix**<br>**2. Calculation of coefficient**<br>Calculating the inducement coefficient values for production, income, value added, indirect tax and employment |
| **Step 3:**<br>**Estimating economic effects** | **1. Estimation of economic effects**<br>Estimating intra- and inter-regional economic effect of smart tourism<br>**2. Proposing Implications** |

**Figure 1.** The research process.

The second step involved constructing the input coefficient matrix ($A$) based on the regional I–O table (see Figure 2) [35]. The Bank of Korea has periodically published an annual inter-regional I–O table. The most current version is the 2013 I–O table, published in 2015 [36]. This table summarizes the transaction information of goods and services between industries that make up the national economy for a certain period (usually one year) [35].

**Figure 2.** The I–O table and the input coefficient matrix ($A$) [35].

The input coefficient in Region 1 ($A_{11}^d$) can be calculated by dividing the intermediate demand of Region 1 ($Z_{11}$) into the total input of the good or service of Region 1 ($X_1$), which is described in Equation (1):

$$A_{11}^d = \frac{Z_{11}}{X_1} \tag{1}$$

The input coefficient from Region 1 to Region $r$ ($A_{r1}^d$) can be calculated as in Equation (2):

$$A_{r1}^d = \frac{Z_{r1}}{X_1} \tag{2}$$

The value-added coefficient in Region 1 ($A_1^v$) be calculated as in Equation (3):

$$A_1^v = \frac{V_1}{X_1} \tag{3}$$

Based on these equations, we constructed the input coefficient matrix ($A$) and the production inducement coefficient matrix. This matrix "represents the direct, indirect, and induced effects throughout the economy, resulting from one unit change in final demand" ([37] p. 598), which is represented as $\left(I - A^d\right)^{-1}$ ($I$ is the unit vector).

Then we calculated the inducement coefficients for the production, income, value-added, indirect tax, and employment for each sector. Table 2 shows the definitions and formulas of each inducement coefficient.

**Table 2.** Definitions and formulas of each coefficient ([28,37]).

| Types of Coefficient | Definition | Formula |
|---|---|---|
| Production | The ripple effect of one unit change in investment on change in business turnover | The column sums of the production inducement coefficient matrix $\left(I - A^d\right)^{-1}$ |
| Income | The ripple effect of one unit change in investment on change in personal income for residents | $A^p \left(I - A^d\right)^{-1}$ $A^p$ is an inducement coefficient matrix for income |
| Value-added | The ripple effect of one unit change in investment on change in value-added | $A^v \left(I - A^d\right)^{-1}$ $A^v$ is an inducement coefficient matrix for value-added |
| Indirect tax | The ripple effect of one unit change in investment on change in indirect tax | $A^t \left(I - A^d\right)^{-1}$ $A^t$ is an inducement coefficient matrix for indirect tax |
| Employment | The number of jobs created from one unit increase in investment | $A^l \left(I - A^d\right)^{-1}$ $A^l$ is an inducement coefficient matrix for employment |

## 4. Analysis and Results

### 4.1. Smart Tourism Industries

As a result of the Delphi rounds, among the 80 sectors of the I–O table, a total number of 15 industries were selected as smart tourism industries (Table 3). The relatedness of each of these industries to smart tourism was evaluated as 4 or more by at least six experts. These industries are various, including tourism-related fields (e.g., Air Freight Services, Restaurant and Accommodation Services, Cultural Services, Sports and Entertainment Services, etc.), ICT-related fields (e.g., Computers and Peripherals, Telecommunications Services, etc.), and other fields (e.g., Research and Development, Wholesale and Retail Service, etc.).

**Table 3.** Selected industries that make up the smart tourism industry.

| Category | Sector | Percentage | 5-Point Scale Score (Mean) |
|---|---|---|---|
| Smart tourism | Computers and Peripherals | 75.0 | 4.17 |
| | Communication, Broadcasting, Video and Sound Equipment | 83.3 | 4.50 |
| | Telecommunications Services | 100 | 4.75 |
| | Broadcasting Service | 91.7 | 4.33 |
| | Information Service | 100 | 4.58 |
| | Supply for Software Development and Other IT Services | 91.7 | 4.50 |
| | Motion Picture and Video Production and Distribution | 66.7 | 4.33 |
| | Research and Development | 50.0 | 4.08 |
| | Wholesale and Retail Service | 91.7 | 4.33 |
| | Road Transport Services | 91.7 | 4.50 |
| | Water Transport Services | 83.3 | 4.08 |
| | Air Freight Service | 91.7 | 4.50 |
| | Restaurant and Accommodation Services | 100 | 4.67 |
| | Cultural Services | 100 | 4.67 |
| | Sports and Entertainment Services | 100 | 4.08 |

We integrated these 15 industries into one sector (smart tourism). As a result, all industries were converted to the 29-sector classification based on previous studies: (1) Agricultural, forestry and fisheries, (2) Mining products, (3) Food products and beverages, (4) Textile and leather products, (5) Wood and paper products, printing, and replication, (6) Petroleum and coal products, (7) Chemicals, (8) Non-metallic mineral products, (9) Primary metal products, (10) Metal products, (11) General machinery, (12) Electrical and electronic equipment, (13) Precision instruments, (14) Transport equipment, (15) Other manufacturing products, (16) Electricity, gas, steam, and air conditioning supply, (17) Water supply, sewerage, waste management, and remediation, (18) Construction, (19) Transportation, (20) Communications and broadcasting (general), (21) Finance and insurance, (22) Real estate and business service (general), (23) Professional, scientific, and technical activities, (24) Administrative and support service activities, (25) Public administration and defense, compulsory social security, (26) Education, (27) Human health and social work activities, (28) Other service activities, and (29) smart tourism.

### 4.2. The Intra- and Inter-Regional Economic Effects

Tables 4 and 5 show the results of the I–O analysis by revealing the mean values of the inducement coefficients for intra- and inter-regional production, income, value-added, indirect tax, and employment of the smart tourism industry and non-smart tourism industry. The sector of the non-smart tourism industry consists of 65 industry sectors that were not identified as smart tourism industries in the Delphi survey.

First, the mean values of the inducement coefficients for production and employment of the smart tourism industry are lower than those of the non-smart tourism industry. Conversely, the mean values of the inducement coefficients for income, value-added and indirect tax are higher than those of the non-smart tourism industry (Production: $M_{smtr}$ = 2.622 vs. $M_{non-smtr}$ = 2.841; Income: $M_{smtr}$ = 0.623 vs. $M_{non-smtr}$ = 0.462; Value-added: $M_{smtr}$ = 1.189 vs. $M_{non-smtr}$ = 0.962; Indirect tax: $M_{smtr}$ = 0.017 vs. $M_{non-smtr}$ = 0.091; Employment: $M_{smtr}$ = 0.013 vs. $M_{non-smtr}$ = 0.014). This result implies that smart tourism has relatively low production and employment effects in all regions and industries, but it is a high income and high value-added industry and greatly contributes to securing tax revenue.

**Table 4.** Estimation of the inducement coefficients of the smart tourism industry and non-smart tourism industries.

| Industry Sector | | Production | | | Income | | |
|---|---|---|---|---|---|---|---|
| | | Intra | Inter | Total | Intra | Inter- | Total |
| 1 | Agricultural, forestry and fisheries | 1.111 | 1.506 | 2.617 | 0.191 | 0.226 | 0.417 |
| 2 | Mining products | 1.225 | 1.464 | 2.689 | 0.225 | 0.224 | 0.449 |
| 3 | Food products and beverages | 1.211 | 1.783 | 2.994 | 0.152 | 0.204 | 0.356 |
| 4 | Textile and leather products | 1.285 | 1.588 | 2.874 | 0.137 | 0.167 | 0.304 |
| 5 | Wood and paper products, printing, and replication | 1.195 | 1.718 | 2.913 | 0.198 | 0.263 | 0.460 |
| 6 | Petroleum and coal products | 1.349 | 1.444 | 2.793 | 0.186 | 0.174 | 0.361 |
| 7 | Chemicals | 1.159 | 1.809 | 2.968 | 0.153 | 0.213 | 0.366 |
| 8 | Non-metallic mineral products | 1.241 | 2.012 | 3.253 | 0.123 | 0.180 | 0.303 |
| 9 | Primary metal products | 1.167 | 2.066 | 3.233 | 0.133 | 0.239 | 0.372 |
| 10 | Metal products | 1.128 | 1.977 | 3.105 | 0.125 | 0.221 | 0.347 |
| 11 | General machinery | 1.160 | 1.955 | 3.114 | 0.171 | 0.268 | 0.439 |
| 12 | Electrical and electronic equipment | 1.094 | 1.682 | 2.777 | 0.158 | 0.240 | 0.398 |
| 13 | Precision instruments | 1.111 | 1.645 | 2.757 | 0.115 | 0.252 | 0.367 |
| 14 | Transport equipment | 1.151 | 2.195 | 3.346 | 0.162 | 0.286 | 0.448 |
| 15 | Other manufacturing products | 1.127 | 1.693 | 2.820 | 0.138 | 0.373 | 0.511 |
| 16 | Electricity, gas, steam, and air conditioning supply | 1.123 | 1.832 | 2.956 | 0.230 | 0.134 | 0.364 |
| 17 | Water supply, sewerage, waste management, and remediation | 1.100 | 1.574 | 2.674 | 0.136 | 0.270 | 0.405 |
| 18 | Construction | 1.241 | 1.931 | 3.172 | 0.253 | 0.385 | 0.638 |
| 19 | Transportation | 1.194 | 1.344 | 2.538 | 0.296 | 0.325 | 0.620 |
| 20 | Communications and broadcasting (general) | 1.168 | 1.501 | 2.669 | 0.315 | 0.337 | 0.652 |
| 21 | Finance and insurance | 1.191 | 1.179 | 2.370 | 0.219 | 0.264 | 0.483 |
| 22 | Real estate and business service (general) | 1.253 | 1.203 | 2.457 | 0.231 | 0.178 | 0.410 |
| 23 | Professional, scientific, and technical activities | 1.459 | 1.258 | 2.717 | 0.370 | 0.426 | 0.797 |
| 24 | Administrative and support service activities | 1.393 | 1.235 | 2.628 | 0.313 | 0.508 | 0.821 |
| 25 | Public administration and defense, compulsory social security | 1.275 | 1.154 | 2.428 | 0.408 | 0.516 | 0.924 |
| 26 | Education | 1.366 | 1.248 | 2.615 | 0.350 | 0.606 | 0.956 |
| 27 | Human health and social work activities | 1.470 | 1.432 | 2.903 | 0.271 | 0.429 | 0.700 |
| 28 | Other service activities | 1.424 | 1.651 | 3.075 | 0.476 | 0.369 | 0.845 |
| **Non-Smart Tourism (mean)** | | **1.197** | **1.644** | **2.841** | **0.197** | **0.264** | **0.462** |
| **29** | **Smart tourism** | **1.234** | **1.388** | **2.622** | **0.382** | **0.241** | **0.623** |

Second, the effects of Seoul smart tourism for income, value-added, indirect tax, and employment in Seoul itself are greater than in other regions (Income: $M_{seoul}$ = 0.382 vs. $M_{other}$ = 0.241; Value-added: $M_{seoul}$ = 0.691 vs. $M_{other}$ = 0.497; Indirect tax: $M_{seoul}$ = 0.009 vs. $M_{other}$ = 0.008; Employment: $M_{seoul}$ = 0.010 vs. $M_{other}$ = 0.003). This result implies that high income, high value-added, and increased tax revenue, which are the strengths of smart tourism, are more pronounced in Seoul. In addition, the employment inducement effects of Seoul within Seoul itself were found to be greater than in other regions.

As for each coefficients' results, the production-inducing effect of smart tourism city Seoul is relatively lower than that of other industries, and has high production ripple effects in other regions. This result implies that Seoul has a weak production base in smart tourism industries, but it contributes to balanced regional development from a macro perspective. The income-inducing effects and value-added-inducing effects of Seoul smart tourism are higher than those of other industries and have high intra-regional effects. This result implies that smart tourism city Seoul creates high income and high value-added. The indirect tax-inducing effects of Seoul smart tourism are higher than in other industries and other regions. Since both intra- and inter-regional effects of Seoul were found to have an effect that exceeded the average value of the indirect tax coefficient (0.007), the creation of smart tourism city Seoul contributes to securing tax revenues for both Seoul and other regions. The employment inducement effects of the smart tourism industries in Seoul were slightly lower than that of other industries, but the effect on Seoul was far greater than that on other regions. This implies that smart tourism city Seoul creates many more jobs for Seoul citizens.

**Table 5.** Estimation of the inducement coefficients of the smart tourism industry and non-smart tourism industries (continued).

| Value Added | | | Indirect Tax | | | Employment | | |
|---|---|---|---|---|---|---|---|---|
| **Intra** | **Inter** | **Total** | **Intra** | **Inter** | **Total** | **Intra** | **Inter** | **Total** |
| 0.606 | 0.693 | 1.299 | 0.042 | 0.043 | 0.085 | 0.021 | 0.004 | 0.025 |
| 0.626 | 0.654 | 1.280 | 0.005 | 0.006 | 0.011 | 0.004 | 0.003 | 0.007 |
| 0.354 | 0.492 | 0.846 | 0.002 | 0.005 | 0.007 | 0.006 | 0.009 | 0.015 |
| 0.327 | 0.376 | 0.703 | 0.002 | 0.002 | 0.005 | 0.007 | 0.005 | 0.011 |
| 0.381 | 0.502 | 0.883 | 0.002 | 0.004 | 0.006 | 0.014 | 0.004 | 0.018 |
| 0.421 | 0.403 | 0.823 | 0.002 | 0.002 | 0.004 | 0.005 | 0.005 | 0.010 |
| 0.313 | 0.446 | 0.759 | 0.002 | 0.003 | 0.005 | 0.008 | 0.003 | 0.012 |
| 0.315 | 0.488 | 0.803 | 0.002 | 0.004 | 0.006 | 0.006 | 0.005 | 0.011 |
| 0.254 | 0.442 | 0.696 | 0.002 | 0.003 | 0.005 | 0.006 | 0.004 | 0.010 |
| 0.239 | 0.423 | 0.662 | 0.002 | 0.003 | 0.005 | 0.007 | 0.003 | 0.010 |
| 0.313 | 0.484 | 0.797 | 0.002 | 0.003 | 0.005 | 0.007 | 0.004 | 0.011 |
| 0.378 | 0.500 | 0.879 | 0.002 | 0.003 | 0.005 | 0.008 | 0.003 | 0.011 |
| 0.251 | 0.490 | 0.741 | 0.002 | 0.003 | 0.004 | 0.007 | 0.003 | 0.010 |
| 0.287 | 0.488 | 0.775 | 0.002 | 0.003 | 0.005 | 0.006 | 0.005 | 0.011 |
| 0.222 | 0.599 | 0.821 | 0.001 | 0.003 | 0.004 | 0.016 | 0.004 | 0.020 |
| 0.393 | 0.544 | 0.936 | 0.002 | 0.002 | 0.004 | 0.007 | 0.001 | 0.008 |
| 0.431 | 0.651 | 1.082 | 0.002 | 0.004 | 0.005 | 0.009 | 0.003 | 0.012 |
| 0.521 | 0.563 | 1.084 | 0.004 | 0.007 | 0.011 | 0.009 | 0.004 | 0.014 |
| 0.393 | 0.617 | 1.010 | 0.006 | 0.004 | 0.011 | 0.012 | 0.004 | 0.015 |
| 0.418 | 0.479 | 0.898 | 0.006 | 0.006 | 0.012 | 0.013 | 0.004 | 0.017 |
| 0.412 | 0.532 | 0.944 | 0.005 | 0.003 | 0.008 | 0.009 | 0.003 | 0.012 |
| 0.489 | 0.685 | 1.174 | 0.006 | 0.016 | 0.021 | 0.009 | 0.002 | 0.010 |
| 0.679 | 0.666 | 1.345 | 0.005 | 0.004 | 0.009 | 0.012 | 0.002 | 0.014 |
| 0.603 | 0.736 | 1.339 | 0.006 | 0.004 | 0.010 | 0.026 | 0.002 | 0.028 |
| 0.679 | 0.829 | 1.507 | 0.005 | 0.000 | 0.006 | 0.010 | 0.001 | 0.011 |
| 0.493 | 0.786 | 1.279 | 0.006 | 0.003 | 0.009 | 0.015 | 0.002 | 0.017 |
| 0.634 | 0.602 | 1.236 | 0.005 | 0.003 | 0.007 | 0.017 | 0.004 | 0.021 |
| 0.637 | 0.658 | 1.295 | 0.004 | 0.005 | 0.009 | 0.025 | 0.003 | 0.028 |
| **0.415** | **0.546** | **0.961** | **0.006** | **0.007** | **0.013** | **0.010** | **0.003** | **0.014** |
| **0.691** | **0.497** | **1.189** | **0.009** | **0.008** | **0.017** | **0.010** | **0.003** | **0.013** |

Figure 3 reveals the ratio of intra- and inter-regional economic effects of the smart tourism sector. Compared to non-smart tourism industries, where more than half of the economic ripple effect is distributed to other regions, the economic effects of smart tourism industries appear to have a lot of influence in Seoul.

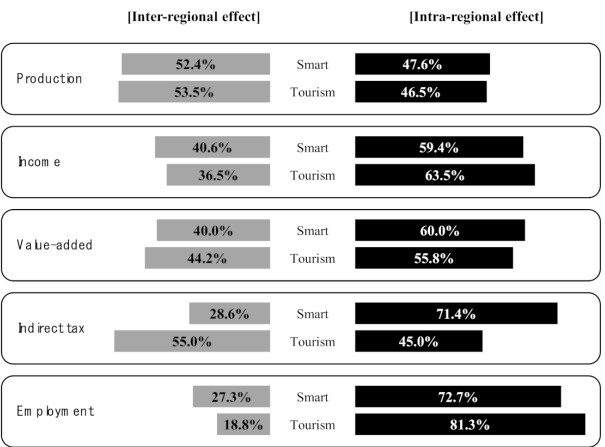

**Figure 3.** Ratio of intra- and inter-regional economic effects of the smart tourism.

### 4.3. Forward and Backward Linkage Effects

For a deeper understanding of the relationships between smart tourism industries and other industries, we conducted a linkage analysis. In the I–O model, an intermediate sector has a relationship with other sectors. Sector *A* supplies input to Sector *B*, and the output of the *B* sector is used as Sector *A*'s input [31]. Based on these relationships, the linkage effect consists of the forward linkage effect and the backward linkage effect. The forward linkage effect refers to "the direct and indirect effects on the production of all other industries that use the output of a specific industry invested as intermediate goods" ([38] p. 2). The backward linkage effect refers to "the direct and indirect effects on the production of all the industries that provide the intermediate inputs necessary for the production of a particular industry being invested in" ([38] p. 2). In other words, the forward linkage effect analysis identifies the output of the smart tourism industry as a raw material for the production of other industries, while the backward linkage effect analysis regards the output of the smart tourism industry as the final product and raw material for the smart tourism industry. The forward linkage (FL) effect and the backward linkage (BL) effect can be defined as the following:

$$FL_i = \frac{\frac{1}{n} \sum_{j=1}^{n} B_{ij}}{\frac{1}{n^2} \sum_{i=1}^{n} \sum_{j=1}^{n} B_{ij}} \tag{4}$$

$$and \; BL_j = \frac{\frac{1}{n} \sum_{i=1}^{n} B_{ij}}{\frac{1}{n^2} \sum_{i=1}^{n} \sum_{j=1}^{n} B_{ij}} \tag{5}$$

where $n$ is the number of industry sectors and $B$ is $\left( I - A^d \right)^{-1}$.

Based on the two linkage effects, industries can be categorized as "intermediate primary", "intermediate manufacture", "final primary production", or "final manufacture" (see Figure 4). "Final primary production" has a low level of forward and backward linkage effects, while "intermediate manufacture" has a high level of both effects. "Intermediate primary production" has a high level of forward linkage effect but a low level of backward linkage effect, while "final manufacture" has a low level of forward linkage effect but a high level of backward linkage effect.

|  | | Forward | |
|---|---|---|---|
| | | Low | High |
| **Backward** | Low | Final primary production | Intermediate primary production |
| | High | Final manufacture | Intermediate manufacture |

**Figure 4.** Industry classification based on linkage effects.

Tables 6 and 7 show the results of the linkage effects in the intra-region and inter-region, respectively. In Seoul, the forward linkage effect of smart tourism is 1.202, which greater than 1 and shows a high level in comparison with other industries (Rank 5). The backward linkage effect (1.234) is also greater than 1 but shows a relatively low level in comparison with other industries (Rank 12). Therefore, in Seoul, the smart tourism industry can be identified as "intermediate primary production". On the other hand, in other regions, the forward and backward linkage effects of smart tourism are found to be low. Therefore, the smart tourism industry in Seoul can be identified as "final primary production" from the perspective of other regions. However, since smart tourism is a combination of cutting-edge technologies and tourism (the service industry), it is not appropriate to refer to it as "primary production" [11]. Therefore, the smart tourism industry can be referred to as "intermediate production" in Seoul and "final production" in other regions, respectively. The intra-regional and inter-regional forward and backward linkages of smart tourism industries and other industries are depicted in Figure 5.

**Table 6.** Forward and backward linkage effects results (intra-regional effects).

| | Industry Sector | Forward Linkage Effect | Ranking | Backward Linkage Effect | Ranking |
|---|---|---|---|---|---|
| 1 | Agricultural, forestry and fisheries | 0.853 | 22 | 1.111 | 26 |
| 2 | Mining products | 0.833 | 29 | 1.225 | 13 |
| 3 | Food products and beverages | 0.865 | 17 | 1.211 | 14 |
| 4 | Textile and leather products | 1.022 | 8 | 1.285 | 7 |
| 5 | Wood and paper products, printing, and replication | 0.921 | 13 | 1.195 | 15 |
| 6 | Petroleum and coal products | 0.850 | 24 | 1.349 | 6 |
| 7 | Chemicals | 0.842 | 27 | 1.159 | 21 |
| 8 | Non-metallic mineral products | 0.859 | 19 | 1.241 | 10 |
| 9 | Primary metal products | 0.893 | 15 | 1.167 | 19 |
| 10 | Metal products | 0.856 | 21 | 1.128 | 23 |
| 11 | General machinery | 0.858 | 20 | 1.160 | 20 |
| 12 | Electrical and electronic equipment | 0.851 | 23 | 1.094 | 29 |
| 13 | Precision instruments | 0.839 | 28 | 1.111 | 26 |
| 14 | Transport equipment | 0.845 | 26 | 1.151 | 22 |
| 15 | Other manufacturing products | 0.940 | 10 | 1.127 | 24 |
| 16 | Electricity, gas, steam, and air conditioning supply | 0.932 | 12 | 1.123 | 25 |
| 17 | Water supply, sewerage, waste management, and remediation | 0.880 | 16 | 1.100 | 28 |
| 18 | Construction | 0.861 | 18 | 1.241 | 10 |
| 19 | Transportation | 1.122 | 7 | 1.194 | 16 |
| 20 | Communications and broadcasting (general) | 0.983 | 9 | 1.168 | 18 |
| 21 | Finance and insurance | 1.403 | 2 | 1.191 | 17 |
| 22 | Real estate and business service (general) | 1.325 | 3 | 1.253 | 9 |
| 23 | Professional, scientific, and technical activities | 1.300 | 4 | 1.459 | 2 |
| 24 | Administrative and support service activities | 1.534 | 1 | 1.393 | 4 |
| 25 | Public administration and defense, compulsory social security | 0.934 | 11 | 1.275 | 8 |
| 26 | Education | 0.848 | 25 | 1.366 | 5 |
| 27 | Human health and social work activities | 0.904 | 14 | 1.470 | 1 |
| 28 | Other service activities | 1.123 | 6 | 1.424 | 3 |
| **29** | **Smart tourism** | **1.202** | **5** | **1.234** | **12** |

**Table 7.** Forward and backward linkage effects results (inter-regional effects).

| | Industry Sector | Forward Linkage Effect | Ranking | Backward Linkage Effect | Ranking |
|---|---|---|---|---|---|
| 1 | Agricultural, forestry and fisheries | 0.736 | 20 | 0.943 | 17 |
| 2 | Mining products | 0.816 | 18 | 0.917 | 19 |
| 3 | Food products and beverages | 1.162 | 8 | 1.117 | 9 |
| 4 | Textile and leather products | 0.852 | 16 | 0.995 | 15 |
| 5 | Wood and paper products, printing, and replication | 1.044 | 11 | 1.076 | 10 |
| 6 | Petroleum and coal products | 2.560 | 2 | 0.905 | 20 |
| 7 | Chemicals | 1.234 | 6 | 1.133 | 8 |
| 8 | Non-metallic mineral products | 1.083 | 9 | 1.261 | 3 |
| 9 | Primary metal products | 2.773 | 1 | 1.294 | 2 |
| 10 | Metal products | 1.451 | 5 | 1.239 | 4 |
| 11 | General machinery | 1.176 | 7 | 1.225 | 5 |
| 12 | Electrical and electronic equipment | 1.035 | 12 | 1.054 | 12 |
| 13 | Precision instruments | 0.861 | 15 | 1.031 | 14 |
| 14 | Transport equipment | 0.976 | 13 | 1.375 | 1 |
| 15 | Other manufacturing products | 1.822 | 4 | 1.061 | 11 |
| 16 | Electricity, gas, steam, and air conditioning supply | 2.338 | 3 | 1.148 | 7 |
| 17 | Water supply, sewerage, waste management, and remediation | 0.792 | 19 | 0.986 | 16 |

**Table 7.** *Cont.*

| | Industry Sector | Forward Linkage Effect | Ranking | Backward Linkage Effect | Ranking |
|---|---|---|---|---|---|
| 18 | Construction | 0.669 | 25 | 1.210 | 6 |
| 19 | Transportation | 1.068 | 10 | 0.842 | 24 |
| 20 | Communications and broadcasting (general) | 0.642 | 27 | 0.940 | 18 |
| 21 | Finance and insurance | 0.686 | 23 | 0.739 | 29 |
| 22 | Real estate and business service (general) | 0.667 | 26 | 0.754 | 28 |
| 23 | Professional, scientific, and technical activities | 0.722 | 21 | 0.788 | 25 |
| 24 | Administrative and support service activities | 0.847 | 17 | 0.774 | 27 |
| 25 | Public administration and defense, compulsory social security | 0.627 | 30 | 0.723 | 30 |
| 26 | Education | 0.629 | 29 | 0.782 | 26 |
| 27 | Human health and social work activities | 0.632 | 28 | 0.897 | 21 |
| 28 | Other service activities | 0.680 | 24 | 1.035 | 13 |
| **29** | **Smart tourism** | **0.793** | **22** | **0.802** | **23** |

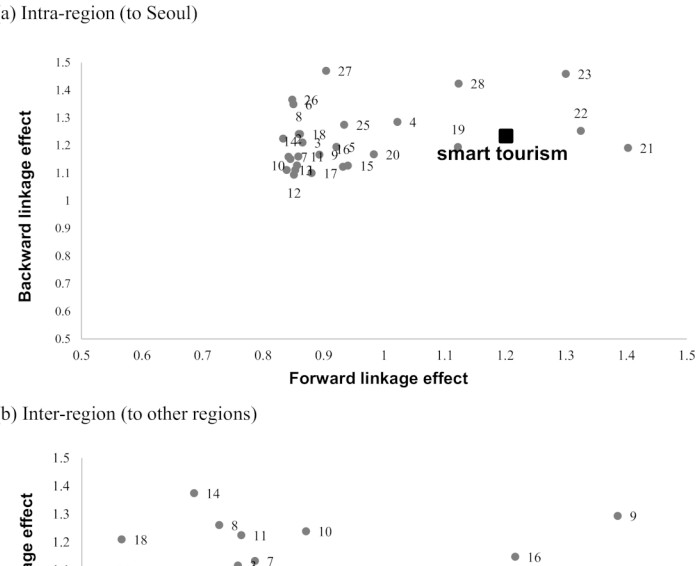

**Figure 5.** Intra- and inter-regional forward and backward linkage effects of smart tourism city Seoul. Note: (1) Agricultural, forestry and fisheries, (2) Mining products, (3) Food products and beverages, (4) Textile and leather products, (5) Wood and paper products, printing, and replication, (6) Petroleum and coal products, (7) Chemicals, (8) Non-metallic mineral products, (9) Primary metal products, (10) Metal products, (11) General machinery, (12) Electrical and electronic equipment, (13) Precision instruments, (14) Transport equipment, (15) Other manufacturing products, (16) Electricity, gas, steam, and air conditioning supply, (17) Water supply, sewerage, waste management, and remediation, (18) Construction, (19) Transportation, (20) Communications and broadcasting (general), (21) Finance and insurance, (22) Real estate and business service (general), (23) Professional, scientific, and technical activities, (24) Administrative and support service activities, (25) Public administration and defense, compulsory social security, (26) Education, (27) Human health and social work activities, (28) Other service activities.

## 5. Discussion and Conclusions

The purpose of this study was to estimate the economic effects and economic roles of smart tourism city Seoul by focusing on the inequalities in geographic and industrial distribution. To achieve the research purposes, we raised two research questions: (1) How much does a smart tourism city contribute to the economies of the city itself and its nearby regions? (2) How does the smart tourism industry have economic relationships with other industries in a smart tourism city? The current study carried out the three steps of analysis. First, we conducted a two-round Delphi survey and identified a total number of 15 industries as smart tourism industries. Then, by adopting the inter-regional I–O model, we investigated the economic effects of smart tourism city Seoul within the city and in other regions. Finally, we calculated the forward and backward linkage effects of the smart tourism industry. As a result, the key findings are as follows:

First, compared to other industries, smart tourism has relatively low production effects in all regions and industries, but it is a high income and high value-added industry and greatly contributes to securing tax revenue. In addition, almost every inducement coefficient of the smart tourism industry in Seoul, except for the production inducement coefficient, is greater than in other regions. This corresponds to the result of a previous study that revealed a weak production inducement effect and strong income, value-added, and indirect tax effects of smart tourism (e.g., [13]). One possible reason is that, due to the nature of the tourism industry, goods and services are not used as raw materials for other industries but are converted into added value [39]. Second, smart tourism city Seoul contributes to the economy of Seoul, but it has relatively small economic effects on other regions than other industries do, except for the indirect tax inducement effect. In addition, in Seoul, the forward linkage effect of the smart tourism sector is higher than the backward linkage effect. From the perspective of Seoul, the smart tourism industry of Seoul is an intermediate production industry, which means that the output of the smart tourism industry is used as an intermediate material for other industries as a kind of "service-type industry" ([40], p. 120). Thus, the smart tourism industry in Seoul has characteristics of the service industry. This result is similar to the result of Shin and Suh's [11] research that revealed a higher forward linkage effect and a lower backward linkage effect of smart tourism. Moreover, this result is similar to that of previous studies on estimating the economic effects of emerging technologies or services, such as Fintech [40] and SmartPort [34].

The approach of the current study—clarifying the scope of smart tourism based on Delphi survey results, estimating the intra- and inter-regional economic effects, and investigating the economic roles of smart tourism—offers theoretical and practical implications.

First, this study tried to estimate the economic effects of smart tourism as objectively as possible. Some previous researchers used arbitrary decisions in order to decide the range of industries relevant to the object to be analyzed. However, this study collected experts' opinions with a two-round Delphi survey and clarified the scope of the smart tourism industry based on the experts' opinions. This contributes to increasing the accuracy of assessing the economic effects and reducing the risk of under- or overestimation. Through this approach, this study could identify the crucial industries related to a smart tourism city. A considerable number of previous studies have demonstrated the definitions, structures, and roles of smart tourism and smart tourism cities (e.g., [4,7,19–22]), but there is a lack of research suggesting smart tourism-related industries. However, this study suggested the 15 industries related to smart tourism and could demonstrate that a smart tourism city can be implemented by the convergence of tourism, ICT, and other numerous industries. Future researchers can replicate the results of the Delphi survey to clarify the scope of smart tourism. Second, this study focused on the intra- and inter-regional economic effects of a smart tourism city. Smart tourism should be implemented at the city level. Therefore, its intra- and inter-regional economic effects should be assessed and compared at the city level as well. However, there is little research to investigate both the intra-and inter-regional

economic effects of smart tourism. This study tried to fulfill this gap via adopting the inter-regional I–O model and providing more useful results for smart tourism city Seoul.

As a practical implication, this study tried to diagnose the economic status of smart tourism city Seoul and provide insight for future direction to policymakers. First, the Delphi survey results showed that a smart tourism city consists of various industrial sectors from broadcasting services to road transport services, which highlights that a smart tourism city is not a city that simply introduces technologies into a tourist city, but a city where tourism, technologies, and numerous industries converge. This study suggests that it is crucial for policymakers to have a smart tourism ecosystem perspective when creating and implementing smart tourism-related policies. Second, it was found that more than half of the production inducement effect of smart tourism city Seoul leaked to other regions. Although the production inducement effect leaked from the smart tourism city to other regions can contribute to balanced regional development, efforts to lay the foundation for the production of a smart tourism city are also needed. Third, the high level of forward linkage effects and the low level of backward linkage effects of the smart tourism industry demonstrated that the main characteristic of smart tourism is the service industry. The output of the service industry is used as intermediate materials for other industries [40]. Therefore, this result highlights the importance of considering the development of other industries directly and indirectly affected by a smart tourism city to maximize the economic effects.

Despite the implications, this study also has limitations. First, this study presented the multipliers of smart tourism rather than the amounts of economic impacts since there is no data of expenditures of tourists or investments from public or private sectors for smart tourism. Therefore, further researchers are asked to estimate the economic effects of a smart tourism city with expenditure or investment data. Second, this study focused on smart tourism city Seoul, but there are other outstanding smart tourism cities, such as Amsterdam, Barcelona, New York City, and so on. Therefore, it will be also meaningful to estimate and compare these cities' economic effects. Finally, although we used the most current version of the I–O table, it was published in 2013. Therefore, the results of this paper are not enough to fully reflect the current economic situation. Further studies, therefore, are required to use an extrapolation method, such as the RAS method developed by Stone [41], which updates the I–O table in different periods based on the existing I–O table [42].

**Author Contributions:** Writing—original draft, H.L.; Writing—review & editing, S.H. All authors have read and agreed to the published version of the manuscript.

**Funding:** This work was supported by the Ministry of Education of the Republic of Korea and the National Research Foundation of Korea (NRF-2019S1A3A2098438).

**Institutional Review Board Statement:** Not applicable.

**Informed Consent Statement:** Not applicable.

**Data Availability Statement:** Not applicable.

**Acknowledgments:** We would like to express sincere gratitude to Chung, N. H.

**Conflicts of Interest:** The authors declare no conflict of interest.

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
