# Peer review of "The Intra- and Inter-Regional Economic Effects of Smart Tourism City Seoul: Analysis Using an Input-Output Model"

_sustainability, doi:10.3390/su13074031_

Round 1
Reviewer 1 Report
The proposed article studies the important question - The Intra- and Inter-Regional Economic Effects of Smart Tourism City. From the overall presentation I would say that an interesting research work has been done. The topic is also important for the readers of the journal. However, I have a few more significant challenges with the paper.
From my point of view, there are some revisions the authors should consider to improve the paper.
- The aim of the paper should be included more clearly in the introduction section.
- I suggest that the authors insert, at the end of the introduction section, a paragraph outlining the layout of the remainder of the manuscript.
- To be consistent with the aims of the Journal, the theme of sustainability must be central or in any case strongly interdependent with the object of the research. Therefore, the authors should include more arguments in order to show the link between the smart tourism city and sustainability.
- The theoretical part remains at a modest level. At this stage, it does not yet provide an in-depth review of the previous literature. It is more a description than analysis. Therefore, a more detailed explanation of theoretical background and research design needs to be supplemented for this paper to be published. The authors should include in this section some research hypotheses.
- The original contribution of the research has to be presented by focusing on the research results based on the research questions.
- The research methods used are appropriate but have limitations, and this should be mentioned. The validation of the models could be presented and justified. Furthermore, the uncertainties of the applied analysis could be discussed. Finally, it would be appropriate to specify in more detail how this research differs from the already published paper that deals with a similar topic.
- The discussion and implications are rather short and they should be extended.
- The quality of the figures is not sufficient.
Revising is always a challenging job, but I think you can develop the paper forward.
Author Response
We sincerely appreciate you, editor, and three anonymous reviewers for providing all valuable comments. As described in this response, we have made significant changes to the paper to address most of the concerns raised. We now believe that our manuscript is much improved after the effort of this revision, and we hope that you concur with our revised paper. Outlined below is a summary of changes made to this revision. We also highlighted the changes within the revised manuscript by using blue-colored texts. The point-by-point responses to the comments will follow immediately after this note. Thank you for your consideration, and we look forward to hearing from you soon.

Reviewer 2 Report
First of all, I am glad to have the opportunity to read your article on “The Intra- and Inter-Regional Economic Effects of Smart Tourism City Seoul: Analysis Using an Input-Output Model”. I would like to say that the paper is interesting in the research design and in the results, but from my humble point of view, I believe that this paper needs revision because the paper has the following relevant weak points:
1) The theoretical background is very limited. A correct study has not been carried out on Smart Tourism City. The study does not provide the most relevant international references in this field. The same problem has the interregional input-output model, where it only offers 6 references. That is, with less than 15 references, develop all the theoretical background. That is, with less than 15 references, all the background is developed, when at least double or triple the references provided would be necessary. Therefore, the literature review is very scarce and completely insufficient.
2) The discussions are very limited. These discussions add almost nothing to academia or management. For this reason, I believe that for the paper to gain quality, the authors must improve them. They must be more profound; they must serve for the advancement in the academy discussing with the contributions of other authors. Surely this could not be done because the theoretical background is very limited in this paper. And it would be the same with discussions for management. They need to be expanded and discussed.
For all these reasons, I think this paper need a revision, but I believe the authors will be able to do it, but they will need to search for relevant references that can better explain the theoretical background and then be able to make a real contribution of their work to the advancement of this field.
Author Response

(The authors gave the same response as above.)

Round 2
Reviewer 1 Report
Dear Authors,
In the revised version, the manuscript has been extended and improved and my comments have been covered.
Best regards
Author Response
Dear reviewer,
We thank you for the reviewer's affirmative feedback.
Best regards
Reviewer 2 Report
I am glad to have the opportunity to read again the article, which has been improved. Nevertheless I consider that the theoretical background is still limited, and more relevant current references should be provided. Finally, in the discussion section I cannot easily see the distinction between the discussions for academia and discussions for management.
Author Response
Dear reviewer,
Thank you for your constructive comments.
We have reviewed the previous literature on smart tourism or smart city, and added the theoretical background and discussion. We hope we could address your concerns in our revised manuscript.
Manuscript Title: The Intra- and Inter-Regional Economic Effects of Smart Tourism City Seoul: Analysis Using an Input-Output Model
We thank the editor and the reviewers for their thoughtful comments and suggestions. The manuscript has been revised based on your comments. We would like to explain how the revision has proceeded for each of your comments. Our responses are highlighted in bold below.
|
Comments |
Response |
Revision in Manuscript |
|
I am glad to have the opportunity to read again the article, which has been improved. Nevertheless I consider that the theoretical background is still limited, and more relevant current references should be provided. Finally, in the discussion section I cannot easily see the distinction between the discussions for academia and discussions for management. |
Thank you very much for your review and feedback. We have reviewed the previous literature on smart tourism or smart city, and divided the smart tourism section into ‘2.1 definition of smart tourism city’ and ‘2.2 economic effect of smart tourism city’. We have also enhanced the conclusion section. We hope we could address your concerns in our revised manuscript.
|
[p. 2] With the explosive growth of technology, the ways of global cities are constructed, consumed and shared have been changed and ‘smartized’ [4,14]. The concept of smart city has been defined by numerous earlier researchers [15-17], but the commonly stated attribute of smart city is that smart city leads efficiency improvement, sustainability, eco-friendliness, and improved residents/tourists quality of life/visit through connectivity through ICTs [4]. At the same time, tourism has been also technologically, economically, and socially developed and ‘smartized’ with the convergence of ICT and tourism [18-19]. The concept of smart tourism put emphasis on achieving symbiosis relation between tourist and citizen and creating economic and social value [18-19]. Therefore, as the term ‘smart’ added to cities and tourism, …
[p. 4, Table 1] Lee et al. (2019): Estimation of the economic effects of tourists’ expenditure in smart tourism city Busan (a city of South Korea)
[p.15] Future researchers can replicate the results of Delphi survey to clarify the scope of smart tourism.
[p.15] This study suggests that it is crucial for policy makers to have a smart tourism ecosystem perspective when creating and implementing smart tourism related policies. |
